# Prognostic Impact of Mucin Expression in Curatively Resected Ampulla of Vater Cancer

**DOI:** 10.3390/cancers16112120

**Published:** 2024-06-01

**Authors:** Byeong Gwan Noh, Hyung Il Seo, Young Mok Park, Su-Bin Song, Suk Kim, Seung Baek Hong, Nam Kyung Lee, Jonghyun Lee, Tae In Kim, Chae Hwa Kwon, Ji Hyun Ahn

**Affiliations:** 1Department of Surgery, Biomedical Research Institute, Pusan National University Hospital, Pusan National University School of Medicine, Busan 49241, Republic of Korea; sagerbk@naver.com (B.G.N.); pym777@hanmail.net (Y.M.P.); songsubin29@gmail.com (S.-B.S.); 2Department of Radiology, Biomedical Research Institute, Pusan National University Hospital, Pusan National University School of Medicine, Busan 49241, Republic of Korea; kimsuk8819@gmail.com (S.K.); cinematiclife7@hanmail.net (S.B.H.); leenk77@hanmail.net (N.K.L.); 3Department of Internal Medicine, Biomedical Research Institute, Pusan National University Hospital, Pusan National University School of Medicine, Busan 49241, Republic of Korea; keiasikr@nate.com (J.L.); zeitgeister88@daum.net (T.I.K.); 4Biomedical Research Institute, Pusan National University Hospital, Busan 49241, Republic of Korea; chkwon@pusan.ac.kr; 5Department of Pathology, Biomedical Research Institute, Pusan National University Hospital, Pusan National University School of Medicine, Busan 49241, Republic of Korea; jvcjh@naver.com

**Keywords:** ampulla of Vater, pancreas, mucins, MUC5AC, overall survival

## Abstract

**Simple Summary:**

It is generally reported that the prognostic factors for ampulla of Vater cancer include T stage, N stage, and lymphovascular invasion and subtype, and the meaning of MUC stain is controversial. This retrospective study by the authors evaluated the significance of various MUC stains in AoV cancer using single-center clinicopathological data. The results showed that MUC5AC was significant for lymph node metastasis and was furthermore valuable as a prognostic factor for predicting overall survival. If this is evaluated in addition to the hematoxylin and eosin stain, which is commonly performed in preoperative biopsy, it is expected to be a method to select groups that require more extensive LN dissection and further prevent locoregional recurrence.

**Abstract:**

*Introduction:* Mucins play a pivotal role in epithelial carcinogenesis; however, their role remains elusive in ampulla of Vater (AoV) cancer, regardless of histological subtype. Therefore, we investigated the clinical significance of MUC1, MUC2, MUC5AC, and MUC6 expression in AoV cancer. *Methods*: Using samples from 68 patients with AoV cancer, we performed immunohistochemical staining for MUC1, MUC2, MUC5AC, and MUC6 using a tissue microarray. Subsequently, we analyzed their expression patterns in relation to clinicopathological parameters and patient outcomes. *Results*: Of the patients, 98.5% exhibited positive expression for MUC1, while MUC2, MUC5AC, and MUC6 were expressed in 44.1%, 47.1%, and 41.2% of the patients, respectively. Correlation analyses between mucin expression and clinicopathological factors revealed no significant associations, except between MUC5AC expression and N stage. Univariate analysis demonstrated significant associations between MUC5AC expression and overall survival (OS). Multivariate analysis further confirmed that MUC5AC expression was a significant predictor of OS, along with the N stage. However, MUC5AC expression was not meaningfully associated with recurrence-free survival (RFS). The patients positive for MUC5AC expression had a considerably shorter OS than those with negative expression. *Conclusions*: Our study provides insights into the clinical impact of mucins on AoV cancer, regardless of the histological subtype. Although MUC1 expression is universal, MUC5AC expression is a significant prognostic indicator that correlates with lymph node metastasis and poor OS. These results emphasize the possible utility of MUC5AC as a biomarker for extensive lymph node dissection and the prognostic evaluation of patients with AoV cancer.

## 1. Introduction

Ampulla of Vater (AoV) cancer is particularly rare, accounting for <1% of gastrointestinal cancers and 6–9% of periampullary malignancies [1,2]. Despite the favorable prognostic indicators associated with AoV cancer, including early detection and high surgical resectability, the outcome can vary significantly depending on its stage at diagnosis. The five-year overall survival (OS) rate for AoV cancer ranges from 20 to 60%, highlighting the impact of disease staging on long-term prognosis [3,4]. 

The histological type of ampullary adenocarcinoma has been independently associated with patient survival and plays an important role in guiding the selection of adjuvant regimens [5,6,7]. Recent reports, including our previous study, have suggested that subtyping using immunohistochemical (IHC) staining alone could have significant clinical implications [8,9,10]. Investigating a combination of different markers and their expression may provide a more comprehensive understanding of tumor biology, which can aid in determining prognosis and tailoring treatment strategies for individual patients.

Of these markers, mucins, which are high-molecular-weight glycoproteins manifested in epithelial tissues, have been explored for their potential influence on carcinogenesis and tumor invasion [11,12]. They are broadly classified into two primary types based on their composition: membrane-bound mucins (MUC1s) and secreted gel-forming mucins (MUC2, MUC5AC, MUC5B, and MUC6). Intestinal-type tumors are typically positive for MUC2 and MUC5AC [8,9] but negative for MUC1 [9]. Pancreatobiliary-type tumors are negative for CDX2, MUC2, and CK20 in general [8,9]. Although the combined detection of mucins has been utilized to distinguish ampullary adenocarcinomas, its clinical significance in AoV cancer remains limited and contradictory [8,9,13,14,15,16]. 

Therefore, our paper aimed to synthetically assess the prevalence of MUC1, MUC2, MUC5AC, and MUC6 expression levels in patients with AoV cancer, their correlation with clinicopathological indicators, and recurrence-free survival (RFS) and OS. 

## 2. Methods

### 2.1. Participants

A retrospective review was conducted of 68 patients diagnosed with adenocarcinoma of the AoV who underwent pancreaticoduodenectomy at Pusan National University Hospital between January 2008 and December 2020. Patients were selected based on a histological slide review by a biliary-pancreatic pathologist. The patients with mucinous, squamous, and neuroendocrine carcinoma; those who underwent R1 resection; exhibited distant metastatic disease, including of the para-aortic lymph nodes; or underwent combined resection of any other organ, including the vessels, were excluded. Tumor staging adhered to the AJCC 8th edition guidelines [17]. 

### 2.2. Follow-Up and Survival Evaluation

Follow-up evaluations were performed on the patients twice a year, which included abdominal and chest computed tomography scans, along with the monitoring of tumor markers (CA19-9 and CEA). RFS, defined as the period between the date of surgery and recurrence, was determined by reviewing the patients’ medical records and imaging study results. OS was calculated as the period between the date of operation and the date of death on the basis of medical records or public administration data, with the endpoint set at 30 November 2023.

### 2.3. Tissue Microarray (TMA) Analysis

Before staining with hematoxylin and eosin (H&E), formalin-fixed, paraffin-embedded tissue samples are created and used to identify appropriate tumor sites by a pathologist specializing in pancreatic diseases. After review, final sampling was conducted. Two representative cores (2.0 mm in diameter) from each tumor were punched and sequentially inserted into the recipient block using TMA. After TMA construction, H&E staining was used to identify the tumor present within the tissue core.

### 2.4. Immunohistochemical Staining 

The immunohistochemical staining of TMA sections was performed using automated staining technology (Benchmark XT; Ventana, Oro Valley, AZ, USA). After deparaffinizing and rehydrating the sections, antigen retrieval was performed and the sections made were MUC1 (Ma695, 1:500; Novocastra Laboratories, Solihull, UK), MUC2 (CLH2, 1:500; Novocastra Laboratories), MUC5AC (CLH5, 1:500) incubated with a primary antibody for 30 min (Novocastra Laboratories), and MUC6 (Ccp58, 1:500; Novocastra Laboratories) using the horseradish peroxidase EnVision system detection kit (DAKO, Glostrup, Denmark) or the ultraVIEW Universal DAB detection kit (Ventana). The reaction was visualized and counterstained with Mayer’s hematoxylin solution.

The immunoreactivities of MUC1, MUC2, MUC5AC, and MUC6 were assessed by a pathologist who was blinded to the clinicopathological data. An immunoreactivity scoring system (IRS) that combines the intensity and proportion of positive cells was used. Each tumor was scored by estimating the average immunoreactivity of the TMA cores. Immunoreactivity was categorized as ‘negative’ (0 to 1 IRS point, no or mild reaction in <10% of positive cells) and as ‘positive’ (>1 IRS point, moderate to intense reaction in ≥10% of positive cells) according to the IRS classification. Representative images of positive immunohistochemical stainig of MUC1, MUC2, MUC5AC, and MUC6 is shown in Figure 1.

### 2.5. Statistical Analysis

The clinicopathological characteristics of AoV cancer patients were analyzed using the Mann–Whitney U test and chi-square test for continuous and categorical variables. The survival comparisons of patients with different mucin expression levels were performed using Kaplan–Meier survival analysis using the log-rank test. Univariate and multivariate analyses were performed using the Cox proportional hazards model. Statistical analyses were performed using the R software (version 4.2.1). Words such as ‘Survival’, ‘Survminer’, and ‘Moonbook’ were used. The cutoff for statistical significance was set at *p* < 0.05. 

## 3. Results

### 3.1. Expression of Mucins in AoV Cancers

Figure 2 and Appendix A show the expression profiles of different mucin types (MUC1, MUC2, MUC5AC, and MUC6) in the 68 patients with AoV cancer. Six (8.8%) and one (1.5%) samples were positive and negative for all four mucins, respectively. While the majority of the patients (67/68, 98.5%) exhibited a positive expression of MUC1, the expression of MUC2, MUC5AC, and MUC6 was observed in 30 (44.1%), 32 (47.1%), and 28 (41.2%) patients, respectively. Of the 32 MUC5AC-expressing tumors, 22 (68.8%) were positive for MUC6.

### 3.2. The Correlation between Mucin Expression and Clinicopathological Factors

Next, we analyzed the correlation between the expression levels of MUC2, MUC5AC, and MUC6 and various clinicopathological factors in the 68 patients with AoV cancer (Appendix A). After excluding MUC1 due to its universally positive expression (excluding one patient), the results revealed no statistically significant correlations between mucin expression and age, sex, T stage, tumor differentiation, lymphovascular invasion (LVI), and perineural invasion (PNI). However, an association was identified between the expression of MUC5AC and the N stage (*p* = 0.017). 

### 3.3. Prognostic Significance of MUC5AC Expression

Univariate analysis revealed significant associations between the T stage (Hazard ratio [HR]: 1.678, confidence interval [CI]: 1.055–2.67, *p* = 0.029), N stage (HR: 2.142, CI: 1.359–3.376, *p* = 0.001), LVI (HR: 2.384, CI: 1.169–4.864, *p* = 0.017), and MUC5AC expression (HR: 2.781, CI: 1.338–5.777, *p* = 0.006) and OS. Multivariate analysis identified the N stage (HR: 1.708, CI: 1.025–2.848, *p* = 0.04) and MUC5AC expression (HR: 2.3, CI: 1.095–4.828, *p* = 0.028) as the significant predictors of OS (Table 1). Univariate analysis indicated associations between the T stage (HR: 2.044, CI:1.164–3.589, *p* = 0.013) and N stage (HR: 1.935, CI: 1.174–3.19, *p* = 0.01), and LVI (HR: 2.783, CI: 1.248–6.205, *p* = 0.012) and RFS, while MUC5AC expression did not exhibit such an association. However, in multivariate analysis, none of the factors, including the T stage, N stage, or LVI, showed a significant association with RFS (Table 2). 

The patients with AoV cancer presenting a positive MUC5AC expression had a notably shorter OS than those with negative expression levels (*p* = 0.004, Figure 3A). The 3- and 5-year OS rates were 51.9% and 30.3%, respectively, in the patients with positive expression, compared to 78.8% and 61.9%, respectively, in those with negative expression. Conversely, RFS did not show significant differences according to the MUC5AC expression levels (*p* = 0.059; Figure 3B).

## 4. Discussion

Abnormal mucin expression has been found in epithelial malignancies [11,12]. Altered mucin expression has been extensively studied to classify histological subtypes in ampullary adenocarcinoma. Among the mucins, MUC1, MUC2, MUC5AC, and MUC6 have been the focus of numerous investigations [13]. In our study, we attempted to explore the prognostic importance of these mucins in AoV cancer, confirming previously known factors, such as T stage, N stage, and lymphovascular invasion, along with identifying MUC5AC as a prognostic factor.

AoV cancer typically exhibits a desmoplastic stroma and commonly expresses MUC1, MUC5AC, MUC6, and MUC2 [13,14]. Consistent with the existing literature, our results demonstrated that MUC1 expression had the highest incidence, with other mucins showing similar frequencies. MUC1, a transmembrane mucin glycoprotein, is commonly detected in epithelial cells and has been widely studied as a tumor marker [11]. Its overexpression contributes to metastasis by inhibiting tumor cell adhesion and allowing the evasion of immune surveillance [11,12]. In AoV cancer, MUC1 expression shows a connection between advanced disease stage and poor prognosis [13]. However, on account of the limited sample size, its clinical role is difficult to generalize. Although MUC1 has also been suggested to distinguish the pancreatobiliary subtypes of ampullary carcinomas, its use has not been independently validated and it mostly has been examined in conjunction with CK or CDX2 [8,9]. Furthermore, the reported positivity rates of MUC1 in AoV cancer vary widely, ranging from 70% to 100% [13,14], complicating its use as a prognostic value. In the present study, the MUC1 expression was 98.5%. Thus, its potential as a prognostic factor or standard for subtype classification has not been adequately assessed. 

Moreover, MUC1 overexpression is associated with drug resistance in various tumors [12,18]. There is still no adjuvant chemotherapy regimen for AoV, which may be due to the resistance of MUC1 to cytotoxic chemotherapy drugs, such as gemcitabine and 5-FU. Therefore, the role of MUC1 in determining treatment strategies requires further investigation. 

MUC2 is primarily expressed in gel-forming goblet cells and has been suggested as a potential biomarker for identifying the intestinal subtype of AoV cancer [8,9]. However, its independent prognostic significance remains unclear, as previous studies have typically described it in conjunction with CK20 or CDX2, without establishing its prognostic role. A study by Santini reported the lack of a prognostic role for MUC2 expression in AoV cancer, albeit with a small sample size [19]. Consistent with this, our findings indicated no correlation between MUC2 expression and OS or RFS in patients with AoV cancer.

Our study highlights MUC5AC as an independent prognostic indicator for AoV cancer. Similar to MUC2, MUC5AC is a gel-forming mucin expressed in gastric foveolar and tracheobronchial epithelial cells. The dysregulation of MUC5AC expression shows varied correlations with survival and prognosis in different cancers. For instance, decreased expression is associated with poor prognosis in gastric carcinoma, whereas increased expression is associated with poor prognosis in pancreatic, colon, and lung cancers [20,21,22,23,24]. However, the prognostic value of MUC5AC in ampullary carcinomas remains controversial. Recent studies by Xue et al. supported MUC5AC as a strong prognosticator [16], whereas Perkins et al. did not observe any prognostic role [25]. Our findings provide limited evidence suggesting that MUC5AC may be a significant prognostic factor for ampullary carcinoma. Further research is necessary to validate the clinical impact of MUC5AC in AoV cancers.

Lymph node metastasis has been identified as a prognostic factor in AoV cancer [15], and our study revealed an association between MUC5AC and lymph node metastasis. Although the mechanism by which MUC5AC affects AoV cancer progression remains unclear, it is believed to regulate cell–cell and cell–stroma interactions, thereby enhancing the invasiveness and metastatic behavior of various cancer types [12]. Sanada et al. showed that MUC5AC expression was higher in the invasive components of ampullary carcinoma, including invasive vascular lesions and lymph node metastases [26]. Similarly, Jun et al. reported that MUC5AC is more commonly expressed in ampullary carcinomas with a higher tumor microenvironment prognostic risk and high invasiveness [27]. These results indicate the involvement of MUC5AC in tumor development, including invasion and metastasis. Importantly, MUC5AC expression emerged as an independent predictor of poor survival in AoV cancer, along with lymph node metastasis, as determined by multivariate analysis. Moreover, our study revealed that MUC5AC expression is significantly related to aggressive clinicopathological features of AoV cancer, such as lymph node metastasis. However, further studies are needed to elucidate the association between MUC5AC and the aggressive behavior of AoV cancer.

Among the periampullary carcinomas, AoV cancer has a relatively favorable prognosis, owing to the high incidence of R0 resection. Our findings suggest the potential of MUC5AC as a biomarker for lymph node metastasis and the prognosis of AoV cancer. The lymphatic drainage of AoV cancer is known to be the first spreading area in the posterior pancreaticoduodenal node group, and is then reported to proceed to the para-aortic area via the inferior pancreaticoduodenal artery area. Hence, the detection of MUC5AC expression in preoperative biopsy specimens may warrant consideration for radical lymph node dissection including the first jejunal branch of the superior mesenteric artery, retroperitoneal and para-aortic node [28]. Additionally, it could aid in the prognostic assessment of AoV cancer. 

MUC6-positive expression reportedly signifies gastric differentiation, while negative expression correlates with the intestinal type of AoV cancer [29]. However, its prognostic significance for AoV cancer remains unclear. Our study did not reveal any association with clinicopathological variables. Interestingly, we observed a higher frequency of the co-expression of MUC5AC and MUC6, although this difference was not statistically significant. A previous study reported that the co-expression of these two mucins is associated with a better prognosis in other types of AoV cancers [29]. Further validation of the correlation between MUC5AC and MUC6 expression in a larger cohort is required.

Despite the inherent limitations of a single-institution study conducted by a single researcher, our findings offer valuable insights into the clinicopathological variations in mucin expression among rare AoV cancer cases, irrespective of the histological subtype. Previous studies, including ours, have established that CK7, CK20, and CDX2 are sufficient markers for subtyping [10]. Therefore, in this study, we did not explore the association between mucin expression and histological subtype. Although the AoV phenotype is defined through immunohistochemical markers in TMA samples using IHC labeling, histochemical testing for all subtypes of mucin is not mandatory [30]. Moreover, the use of TMA may affect MUC expression as it may be heterogeneous except for MUC1, which shows ubiquitous expression. We focused on the clinical significance of mucins, particularly their prognostic value in AoV cancer. Moreover, it is essential to note that the threshold for mucin expression used in our study was set at >1 IRS point, with no or mild reaction in <10% of the positive cells. Another study on ampullary carcinoma, which set the cutoff for MUC5AC positivity at either >25% or 1%, reported significant survival differences between MUC5AC-positive and MUC5AC-negative patients, regardless of the histological subtype [16]. Thus, despite the differences in cutoff values, the detection of any positive expression of MUC5AC in patients with AoV cancer is associated with worse survival outcomes.

## 5. Conclusions

In conclusion, our findings provide valuable insights into the clinicopathological differences in mucin expression in AoV cancer regardless of the histological subtype. While MUC1 expression is ubiquitous, MUC5AC expression has emerged as a significant prognostic factor associated with lymph node metastasis and poor OS. Therefore, MUC5AC expression in preoperative biopsies can serve as a useful biomarker for extensive lymph node dissection and prognostic evaluation in patients with AoV cancer. Further validation and mechanistic exploration are warranted to elucidate the clinical impact of mucin expression in AoV cancer and pave the way for personalized therapeutic approaches.

## Figures and Tables

**Figure 1 cancers-16-02120-f001:**
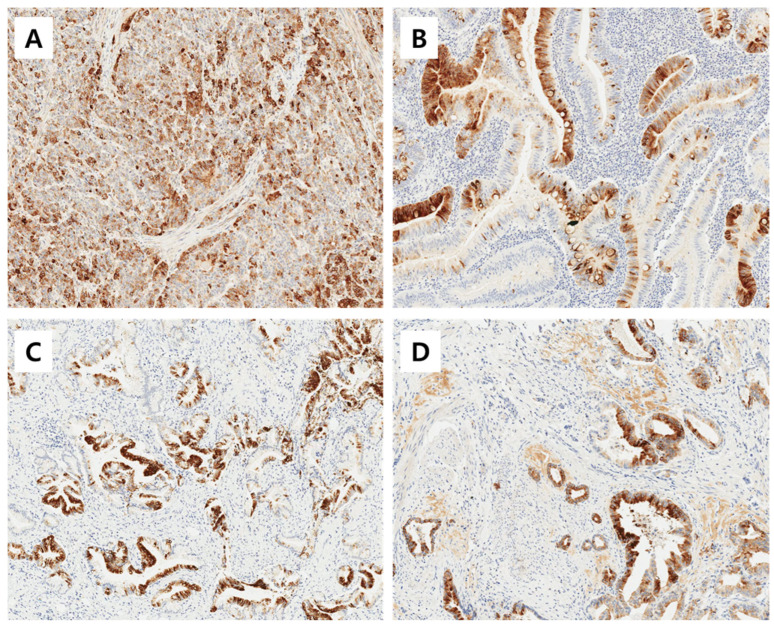
Representative images of positive immunohistochemical stainig of MUC1 (**A**), MUC2 (**B**), MUC5AC (**C**), and MUC6 (**D**). Magnification ×200.

**Figure 2 cancers-16-02120-f002:**
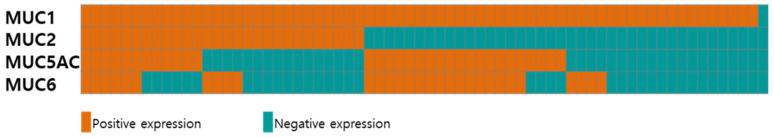
Expression of mucins in 68 patients with AoV cancer. Tumors with positive and negative expression are presented in red and blue, respectively.

**Figure 3 cancers-16-02120-f003:**
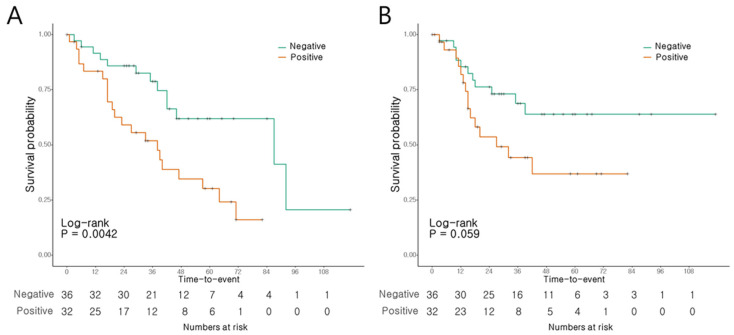
Survival of AoV cancer patients according to MUC5AC expression. (**A**) Overall survival. (**B**) Recurrence-free survival.

**Table 1 cancers-16-02120-t001:** Univariate and multivariate analysis for overall survival of 68 AoV cancer patients.

	Univariate		Multivariate	
	HR (CI)	*p*-Value	HR (CI)	*p*-Value
Age (≥65)	1.379 (0.664–2.865)	0.389		
Sex (male)	1.117 (0.52–2.401)	0.777		
Differentiation (poor)	1.106 (0.506–2.419)	0.8		
T	1.678 (1.055–2.67)	0.029	1.303 (0.778–2.184)	0.215
N	2.142 (1.359–3.376)	0.001	1.708 (1.025–2.848)	0.04
LVI	2.384 (1.169–4.864)	0.017	0.328 (0.582–3.03)	0.499
PNI	0.949 (0.474–1.899)	0.882		
MUC2	0.587 (0.28–1.23)	0.158		
MUC5AC	2.781 (1.338–5.777)	0.006	2.3 (1.095–4.828)	0.028
MUC6	1.202 (0.612–2.363)	0.593		

**Table 2 cancers-16-02120-t002:** Univariate and multivariate analysis for recurrence-free survival of 68 AoV cancer patients.

	Univariate		Multivariate	
	HR (CI)	*p*-Value	HR (CI)	*p*-Value
Age (≥65)	1.189 (0.539–2.626)	0.668		
Sex (male)	1.044 (0.439–2.486)	0.922		
Differentiation (poor)	1.915 (0.85–4.312)	0.116		
T	2.044 (1.164–3.589)	0.013	1.622 (0.873–3.013)	0.126
N	1.935 (1.174–3.19)	0.01	1.351 (0.74–2.464)	0.327
LVI	2.782 (1.248–6.205)	0.012	1.717 (0.678–4.351)	0.254
PNI	0.903 (0.409–1.991)	0.8		
MUC2	0.854 (0.387–1.883)	0.695		
MUC5AC	2.086 (0.955–4.555)	0.065		
MUC6	0.813 (0.369–1.793)	0.608		

## Data Availability

The data that support the findings of this study are available from the corresponding author.

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
