# Peer review of "Prognostic Impact of Mucin Expression in Curatively Resected Ampulla of Vater Cancer"

_cancers, 2024, doi:10.3390/cancers16112120_

Round 1

Reviewer 1 Report

Comments and Suggestions for Authors

The authors present a very nice study on Ampullary adenocarcinoma. It is a rather rare tumor and their cohort has good numbers. Although the WHO classification prompts for MUC staining to achieve a proper classification, this is not always the rule. This work will stress this factor. 

This should be stressed in the discussion. Also, the use of TMA may influence the MUC expression since it could be heterogeneous, except for MUC1 - this should be stated as a limitation (although TMA use is now a current and valid methodology)

The authors should make clearer the significance of "extensive lymph node dissection", especially in the abstract. These tumours are usually removed by dueodenopancreatectomy. How can the surgeon be more aggressive?

The rationale is solid and the conclusion supports the findings. Figures and tables are of good quality. References are ok

Author Response

Reply to Reviewers’  Comments

 We are grateful to the Editors and Reviewers for meticulous examination and appropriate comments for our manuscript, which has served to ameliorate this revision. We have amended our manuscript and performed additional analyses to address the Reviewers’ comments and concerns. Detailed point-by-point responses are as follow,

Reviewer 2 Report

Comments and Suggestions for Authors

Byeong Gwan Noh et al present a well-design, executed and written study on the potential role of different mucins expressed in a homogenous sample of ampulla of Vater adenocarcinomas excluding patients with squamous, mucinous, and neuroendocrine carcinomas; R1 resections; distant metastatic disease; vascular reconstruction and/or multiorgan resection. All tumors but one expressed MUC1, therefore MUC2, MUC5AC and MUC6 were analyzed. MUC5AC was shown to be associated with N+ disease and overall survival, but not disease-free survival.

I have only a small number of minor observations:

- The title should be revised. Even when MUC5AC could be analyzed in preoperative endoscopic biopsies, lymph node dissection should be routinely performed in all patients. This fact does not reduce the importance of the findings of the study.

- Table 1 adds very little to the manuscript and it can be included as supplementary material. The Results section 3.2 already establishes that there is only one significant association (MUC5AC and LN status).

- There is no correlation analysis performed in the study (neither Pearson nor Spearman). Please review the manuscript and substitute the term with "association".

Congratulations to the authors for this manuscript, it was my pleasure reading and learning from it. 

Author Response

(The authors gave the same response as above.)
